# Triangle Matters! TopDyG: Topology-aware Transformer for Link Prediction on Dynamic Graphs

Submission Id: 1108

## ABSTRACT

Dynamic graph link prediction is widely utilized in the complex web of the real world, such as social networks, citation networks, recommendation systems, etc. Recent Transformer-based link prediction methods on dynamic graphs not only fail to model the fine-grained structures such as **triangles** with the vanilla Transformers in the **graph serialization** process, but also amplify the imbalanced distribution of graphs because of their **over-estimation of high-degree nodes**. To tackle these issues, we propose a Topology-aware Transformer on Dynamic Graph (TopDyG) for link prediction, consisting of a topology injected Transformer (Ti-Transformer) and a mutual information learning (Mi-Learning). The Ti-Transformer explores the explicit structure of serialized graphs, capturing the topological features. The Mi-Learning mines the relationship between nodes by modeling the mutual information with a prior knowledge, alleviating the over-estimation of high-degree nodes when applying the Transformer-based models for the dynamic graph link prediction task. Extensive experiments on four public datasets containing both transductive and inductive settings present the superiority of our proposal. In particular, TopDyG presents an improvement of **43.27%** and **28.75%** against the state-of-the-art baselines in terms of NDCG and Jaccard, respectively. The advantages are especially obvious on the high-density graphs. [1]

## CCS CONCEPTS

• **Information systems** → **Web mining**.

## KEYWORDS

Dynamic graphs, Transformer, Link prediction, Topology

**ACM Reference Format:**

Anonymous Author(s). 2025. Triangle Matters! TopDyG: Topology-aware Transformer for Link Prediction on Dynamic Graphs. In *Proceedings of ACM Web Conference 2025 (WWW '25)*. ACM, New York, NY, USA, 10 pages. https://doi.org/XXXXXXX.XXXXXXX

## 1 INTRODUCTION

Graph structure is leveraged for representing various kinds of data on World Wide Web, such as citation networks [4, 14], social networks [11, 45], and recommendation systems [3, 27]. Realistically,

[1] Our code is available at: https://anonymous.4open.science/r/TopDyG-924B.

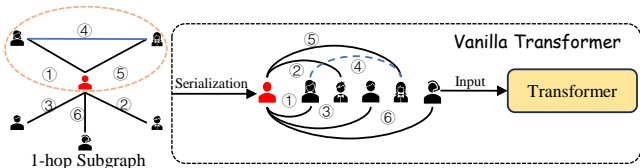

**Figure 1: Graph serialization in vanilla transformer.**

the graph data in the above scenarios is not always static, but continuously evolving over time, i.e., the edges between nodes or the nodes themselves may appear or disappear over time. In reality, forecasting the changes in interactions between nodes on dynamic graphs, such as product clicks in e-commerce, social media followings, or mutual citations in citation networks, is quite prevalent and can serve as a fundamental task. Therefore, we concentrate on the task of link prediction on dynamic graphs, in the hope of capturing the potential pattern to support real-world applications, e.g., knowledge graph completion [26] and social analysis [35].

Recent dynamic graph link prediction works can be mainly divided into two categories: GNN-based methods and Transformer-based methods. The former approaches usually consist of two main modules, i.e., the structural feature extractors like graph neural networks (GNN) [41, 42] and the temporal feature extractors like recurrent neural networks [17] as well as the attention modules [33]. In addition, Cong et al. [4], Wu et al. [40], Yu et al. [44] propose to leverage the Transformer architecture to model the temporal graphs. They creatively convert the initial graphs into node sequences, which help model the structural and temporal relations between nodes. Such paradigm presents a better capability on simulating the long-term temporal dependencies than the GNN-based methods, achieving the state-of-the-art performance.

Albeit much progress, the Transformer-based methods still exhibit two inherent flaws when applying to the dynamic link prediction. The first flaw is **graph serialization**. As a naturally suited model for Euclidean data, the vanilla Transformer can easily handle text [2, 8], images [9] and videos [38]. As for the graph, the non-Euclidean data has to be serialised in an occurrence order before feeding into Transformer [40, 44] (see the upper right part of Fig. 1). As shown in Fig. 1, each graph node as the centre only accesses its neighbors in this way, which can be mapped into a *radial* structure consisting of edges 1, 2, 3, 5, 6, resulting in a missing edge 4. Revisiting the topology of 1-hop subgraph in Fig. 1, we argue that it is **the graph serialization** that destructs the **triangle** relations formed by edges 1, 4 and 5. Analogous to the real world, the triangular relationship has always been an important factor in maintaining the clique stability [24].

To quantify the significance of triangles within the graph structure, we collect the statistics related to the triangles from several realistic datasets in Fig. 2. As shown in Fig. 2a, each node contributes to the formation of at least 1 triangle on average, and nearly half

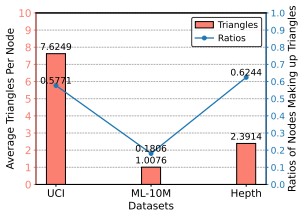

(a) Average number of triangles each node participates in, and ratio of nodes making up triangles.

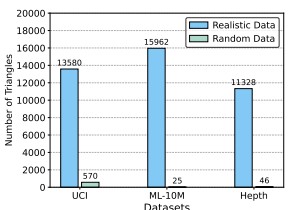

(b) Number of triangles of the randomly generated data and the realistic data.

Figure 2: Triangle-related statistics of three realistic datasets.

nodes are involved in the constructing triangles, except in ML-10M. To overcome the random data noise in realistic graphs, we further employ the Erdős-Rényi algorithm [10] to generate the corresponding random graphs for each dataset based on the number of nodes and edges, and plot their triangle numbers in a pair way in Fig. 2b. Clearly, on three datasets, the number of triangles in the realistic dataset is significantly greater than that in the synthetic one. The above phenomena prove that the triangles are ubiquitous in dynamic graphs. Therefore, modeling the triangles we argue is a key role in capturing the commonalities between neighbors in a 1-hop subgraph [24] and further compensating the vanished edge in graph serialization.

The second flaw in Transformer is **over-estimation of high-degree nodes**. The conditional likelihood in Transformer always make predictions in the maximum-probability way [30]. When it comes to the sequence prediction task, the longer sequence causes the accumulation of more high-probability nodes, e.g., *Hubs* nodes in the scale-free networks. Adding insult to injury, the graph serialization in Transformer converts the link prediction task into a sequence forecasting one. Additionally, most realistic networks, e.g., citation networks and social networks, are scale-free [1], where the node degrees are subject to a power-law distribution. Hence, when the Transformer-based models are trained on the scale-free graphs, they are inclined to predict nodes with extremely high degrees [1], rather than the high-proposition but low-degree nodes.

In this paper, we attempt to provide solutions to above-mentioned issues by proposing a topology-aware Transformer architecture for dynamic graph link prediction, termed **TopDyG**, which consists of two main components: the topology-injected Transformer (Ti-Transformer) and the mutual information learning (Mi-Learning). In particular, for the flaw of graph serialization, the Ti-Transformer explores the structural features, especially triangles, by a simple yet efficient way without any extra trainable networks. For the flaw of over-estimation of high-degree nodes, inspired by the researches on long-tail or class-imbalanced problems [6, 20–22], the Mi-learning obtains the intrinsic correlation of nodes in sequences rather than the high-frequency tokens (*Hubs*), which effectively alleviates the bottleneck caused by the accumulation of high-probability nodes.

We conduct extensive experiments on four public datasets, including three non-bipartite graph datasets rich in triangles and a bipartite graph dataset devoid of triangles. TopDyG outperforms the state-of-the-art baselines on all datasets in terms of NDCG and Jaccard. Specifically, the experimental results illustrate that the advantage of TopDyG over baselines is positively correlated with

the topological statistics, e.g., the graph density and the average number of triangles each node participates in. The contributions of our work can be condensed into the following three aspects.

(1) We give the ability of capturing the explicit topology feature in **serialized dynamic graphs** to Transformer without attaching any trainable module and complex components.
(2) We introduce a novel learning strategy based on mutual information to alleviate the effects caused by **over-estimation of high-degree nodes** in dynamic graphs, thus assisting the model to capture the intrinsic patterns.
(3) We evaluate our proposals on extensive experiments on four real-world datasets and find that TopDyG achieves obvious improvements over several competitive baselines.

## 2 RELATED WORK
### 2.1 Dynamic link prediction

Recent dynamic graph researches can be classified into discrete-time approaches and continuous time ones [7, 46].

For the discrete-time approaches [29, 33, 47], their time sets are discrete, where the events on dynamic graphs between time points are not be recorded completely. Computationally, such models assuming discrete-time domain are easier to manipulate. For instance, one of the representative methods is DySAT [33], which leverages a graph attention network and the self-attention module as cornerstones, aiming to model the structural and temporal features, respectively. Additionally, EvolveGCN combines graph convolution networks with GRUs or LSTMs to learn both structural and temporal features. For the continuous-time approaches [5, 17, 32, 36, 39, 42], they deal with data that fully records the events on the graphs and corresponding timestamps. Therefore, they can capture more details compared to discrete-time methods. Specifically, they always pay more attention to time encoding because of the more abundant temporal resources compared to discrete settings. Wen and Fang [39] propose to integrate both the event and node dynamics to respectively capture the individual and collective features, and Cong et al. [5] design a GNN-free architecture termed GraphMixer with MLPs as well as an offline time encoding function, aiming to capture temporal information. Furthermore, Wu et al. [40] introduce a simple but effective Transformer-based method to predict the future links with serialized subgraphs, which are processed by the temporal alignment technique.

### 2.2 Graph Transformers

Transformers for graph data are a recent advancement in graph data mining, offering a new type of neural network models for graph data [34]. Inspired by the success of transformers in NLP and CV, researchers attempt to integrate them into GNNs to develop their potential in modeling graph structures. For instance, Ying et al. [43] propose Graphormer framework, which is built upon the standard Transformer architecture and can obtain satisfying capability of graph representation with different granularity of structural encoding functions. And some methods attempt to obtain global topology by eigenvectors or eigenvalues of a matrix representation, such as adjacency or Laplacian matrix [16, 31]. Additionally, Min et al. [25]

introduce a graph masking attention mechanism to assist Transformers to capture the graph-related knowledge for modeling the structural features.

When it comes to dynamic graphs, Cong et al. [4] propose a novel but complex architecture termed DyFormer, aiming to capture co-occurrence neighbors of different nodes and encode temporal information. On the other hand, SimpleDyG [40] is a more concise method for dynamic graph than previous researches since that it only rely on the standard Transformer architecture without any learnable attachment, preventing the complex modifications.

The differences between our proposals and the current dynamic graph modeling methods can be summarized in the following folds. First, although our proposals still rely on the vanilla Transformer architecture, we inject the topological information to guide the model capture the graph structure in an explicit way. Second, we modify the optimization goal of Transformers from maximizing the conditional probability of the generated sequence to the mutual information between nodes and their history, considering the prior structure to alleviate the imbalanced distribution problem.

## 3 PRELIMINARY

### 3.1 Problem definition

Formally, a dynamic graph $\mathcal{G}$ can be defined as $\mathcal{G} = \{\mathcal{V}, \mathcal{E}, \mathcal{X}\}$, where $\mathcal{V} = \{v_i, i \in (1, 2, \cdots, |\mathcal{V}|)\}$ is the set of nodes $v_i$ appeared in $\mathcal{G}$; $\mathcal{E} = \{e_{ij} | e_{ij} = (v_i, v_j, t_\tau), \tau \in [1, 2, \cdots, \mathcal{T}]\}$ is the edge set, whose triplets $(v_i, v_j, t_\tau)$ represent an edge $e_{ij}$ connecting nodes $v_i$ and $v_j$ at timestamp $t_\tau$; $\mathcal{X} = [x_1, x_2, \cdots, x_{|\mathcal{V}|-1}, x_{|\mathcal{V}|}] \in \mathbb{R}^{|\mathcal{V}| \times d}$ denotes the $d$-dimension feature set matching with the node set $\mathcal{V}$. Given a dynamic graph $\mathcal{G}$, our task is training a model $f(\mathcal{G}; \theta)$ with parameter $\theta$, to obtain the pattern of temporal evolution of $\mathcal{G}$ at the next timestamp, so that predicting the new edge set $\mathcal{E}[\ ;\ ; t_{\mathcal{T}+1}]$ at the future timestamp $t_{\mathcal{T}+1}$:

$$\begin{cases} \theta^* &= \arg\min_{\theta \in \Theta} \sum_{\tau=2}^{\mathcal{T}} \| \mathcal{E}[\ ;\ ; t_\tau] - f(\mathcal{G}_{t_{\tau-1}}, \theta) \|_F \\ \mathcal{E}[\ ;\ ; t_{\mathcal{T}+1}] &= f(\mathcal{G}_{t_{\mathcal{T}}}, \theta^*) \end{cases}, \quad (1)$$

where $\mathcal{G}_{t_\tau}, \mathcal{E}[\ ;\ ; t_\tau]$ denote the dynamic graph $\mathcal{G}$ and the edge st at timestamp $t_\tau$, respectively; $\Theta$ denotes a parameter space named hypothesis space, and $\|\cdot\|_F$ represent the Frobenius norm of matrix.

### 3.2 Serialization of dynamic graphs

The original graph cannot directly be fed into Transformer-based models since that Transformer is designed for dealing with the Euclidean structure data such as texts [2, 8], images [9] and videos [38]. Vanilla Transformer cannot understand the topology of graph data, therefore, traditional solution is to convert graph data into sequence data for the Transformer to understand and utilize [40].

For example, recent study segments the given dynamic graph into 1-hop subgraphs of each node and transforms the subgraphs into token sequences sorted by interaction time. Suppose the node $v_i$ as the center, its serialized sequence can be represented as:

$$\begin{cases} [v_i, v_i^{t_{1,v_i}}, v_i^{t_{2,v_i}}, \cdots, v_i^{t_{j,v_i}}, \cdots, v_i^{t_{n_i,v_i}}], \\ s.t. \quad \{t_{1,v_i}, t_{2,v_i}, \cdots t_{n_i,v_i}\} \subseteq \{t_1, t_2, \cdots, t_{\mathcal{T}}\} \end{cases} \quad (2)$$

where $v_i^{t_{j,v_i}}$ denotes the node connected with center $v_i$ at timestamp $t_{j,v_i}$. Furthermore, to assist Transformers understand a global time

domain across sequences and synchronize the time interval, the timestamp sequence of center $v_i$ connection can be averagely cut into $T$ subsequences with $\lceil \mathcal{T}/T \rceil$ interval. Correspondingly, the serialized sequence in Eq. (2) is also segmented into subsequences:

$$\begin{cases} S_i^m = \ [v_i^{t_{m,1}}, v_i^{t_{m,2}}, \cdots, v_i^{t_{m,|S_i^m|}}], \\ s.t., \quad [t_{m,1}, t_{m,|S_i^m|}] \subseteq ((m-1)\lceil \mathcal{T}/T \rceil, m\lceil \mathcal{T}/T \rceil ], \\ m \in 1, 2, \cdots, T \end{cases} \quad (3)$$

To assist Transformer to capture the ego node and its temporal pattern, some special tokens are inserted, including the start tokens for historical and predicted data $\langle |hist| \rangle$ and $\langle |pred| \rangle$, and corresponding end tokens $\langle |endofhist| \rangle$ and $\langle |endofpred| \rangle$. Besides, $\langle |time\ m| \rangle$ denotes the begin of $m$-th time period. Hence, the input and output sequence for center node $v_i$ can be represented as:

$$\mathbf{x}_i = [\langle |hist| \rangle, v_i, \langle |time1| \rangle, S_i^1, \langle |time2| \rangle, \cdots, S_i^{T-1},$$
$$\langle |endofhist| \rangle], \quad (4)$$
$$\mathbf{y}_i = [\langle |pred| \rangle, \langle |timeT| \rangle, S_i^T, \langle |endofpred| \rangle].$$

## 4 APPROACHES

### 4.1 Overview

In this section, we detail the TopDyG model, which consists of two main components, i.e. topology-injected Transformer (see Sec. 4.2), and mutual information learning (see Sec. 4.3).

The workflow of TopDyG is plotted in Figure 3. First, we extract the 1-hop subgraph for each node from the original dynamic graph and serialize them into sequences consisting of node tokens and special tokens. Then we capture the topology feature of each sequence by constructing its corresponding normalized adjacent matrix. After that, we inject the topology features into the vanilla mask matrices of Transformer model to shape the **topology-injected Transformer**, which is capable of eliciting the hidden topology information from generated representations. Finally, we utilize the distribution of degree of each node, obtained from the original graph, as the prior knowledge to model the mutual information between tokens and their history, alleviating the over-estimation bottleneck in high-degree nodes.

### 4.2 Topology-injected Transformer

*4.2.1 **Capturing the topology of input**.* Given a dynamic graph $\mathcal{G} = \{G_1, G_2, \cdots, G_{|\mathcal{V}|}\}$ segmented by nodes, where $G_i = \{V_i, E_i, X_i\}$ denotes the 1-hop subgraph centered around the node $v_i$ extracted from $\mathcal{G}$. And $V_i, E_i, X_i$ represent the node set, edge set, and node features of $G_i$, respectively.

For each subgraph $G_i$, we can construct the serialized input sequence $\mathbf{x}^{(i)}$ according to Eq.(4) and abstract it as:

$$\mathbf{x}^{(i)} = [h_1^{(i)}, h_2^{(i)}, \cdots, h_n^{(i)}] \in \mathbb{R}^n, \quad (5)$$

where $\mathbf{x}^{(i)}$ consists of $n$ token ids from common node tokens or special tokens mentioned in Section 3.2.

And then, we obtain the complete binary relation matrix $A^{(i)} = \{0, 1\}^{n \times n}$ of the input sequence $\mathbf{x}^{(i)}$, which not only includes the relationships between the neighbor nodes and the ego node but also the relationships between the neighbor nodes themselves, i.e., $A_{pq}^{(i)} = 1$ if $(h_p^{(i)}, h_q^{(i)}) \in E_i$, otherwise $A_{pq}^{(i)} = 0$. Thus, $\mathbf{x}^{(i)}$ can be

Figure 3: The workflow of TopDyG

regarded as special version of serialized subgraph $G_i$ appending with special tokens.

Finally, we compute the normalized relation matrix $\hat{A}^{(i)}$ of input sequence $\mathbf{x}^{(i)}$ [15]:

$$\hat{A}^{(i)} = D^{(i)-\frac{1}{2}} A^{(i)} D^{(i)-\frac{1}{2}} \in \mathbb{R}^{n \times n}, \tag{6}$$

where $D^{(i)}$ is the degree matrix of $\mathbf{x}^{(i)}$ and $D^{(i)}{}_{pp} = \sum_q A^{(i)}_{pq}$. Compared to the original matrix $A^{(i)}$, the normalized $\hat{A}^{(i)}$ reduces the weights of nodes with high degrees through the $D^{(i)-\frac{1}{2}}$, assisting the model to capture the informative pattern in $\mathbf{x}^{(i)}$.

*4.2.2 **Injecting the topology into Transformer**.* We use $\mathbf{h} \in \mathbb{R}^d$ to represent the $d-$dimensional hidden feature of token $h$ in Eq.(5). Thus, the input sequence $\mathbf{x}^{(i)}$ can be converted to a matrix $\mathbf{H}^{(i)}$:

$$\mathbf{H}^{(i)} = [\mathbf{h}_1^{(i)}, \mathbf{h}_2^{(i)}, \cdots, \mathbf{h}_n^{(i)}] \in \mathbb{R}^{n \times d}. \tag{7}$$

The core component in Transformer is multi-head self-attention module ATTENTION($\cdot$).

Firstly, it encodes $\mathbf{H}^{(i)}$ to a triplet $(\mathbf{Q}^{(i)}, \mathbf{K}^{(i)}, \mathbf{V}^{(i)})$:

$$\mathbf{Q}^{(i)} = \mathbf{H}^{(i)} \mathbf{W}_Q, \mathbf{K}^{(i)} = \mathbf{H}^{(i)} \mathbf{W}_K, \mathbf{V}^{(i)} = \mathbf{H}^{(i)} \mathbf{W}_V, \tag{8}$$

where $\mathbf{W}_Q \in \mathbb{R}^{d \times d_Q}$, $\mathbf{W}_K \in \mathbb{R}^{d \times d_K}$, $\mathbf{W}_V \in \mathbb{R}^{d \times d_V}$ are trainable weights and $d_Q = d_K = d_V = \frac{d}{N_{hd}}$. $N_{hd}$ denotes the number of operations in the multi-head self-attention module, resulting in $(\mathbf{Q}_j^{(i)}, \mathbf{K}_j^{(i)}, \mathbf{V}_j^{(i)})$ for $1 \le j \le N_{hd}$.

Since that we concentrate on the prediction capability for future connected nodes, we decide to utilize the decoder-only architecture to simulate the scenario where only historical nodes are available in practical applications. Therefore, a unit lower triangular mask matrix $M \in \mathbb{R}^{n \times n}$ is utilized for preserving the historical tokens and masking the future tokens:

$$M = \begin{bmatrix} 1 & & \\ \vdots & \ddots & \\ 1 & \cdots & 1 \end{bmatrix}_{n \times n}, \tag{9}$$

where elements with a value of 1 indicate that the token will be retained. Thus, each token in an input sequence can only see the current moment and historical tokens without knowing about future tokens, which models the temporal information. Similarly, we fuse the structural information into the original mask matrix to form the **topology-injected mask matrix** $M^{(i)*}$ as follows:

$$M^{(i)*} = M + \hat{A}_{lt}^{(i)}, \tag{10}$$

where $\hat{A}_{lt}^{(i)}$ denotes the lower triangular part of $\hat{A}^{(i)}$, s.t., $\hat{A}_{lt}^{(i)} = M \odot \hat{A}^{(i)}$. And $\odot$ denotes Hadamard product operation. With such topology-injected mask matrix $M^{(i)*}$, the output of Transformer can be computed as:

$$head_j^{(i)} = \text{SOFTMAX}\left(\frac{\mathbf{Q}_j^{(i)} \mathbf{K}_j^{(i)\top}}{\sqrt{d_K}} \odot \left(M^{(i)*} - \left(\lim_{C \to \infty} C\left(\mathbb{1} - M^{(i)*}\right)\right)\right)\right) \mathbf{V}_j^{(i)}$$

$$\text{ATTENTION}(\mathbf{H}^{(i)}) = \text{CONCAT}(head_1^{(i)}, \cdots, head_{N_{hd}}^{(i)}) \mathbf{W}_O, \tag{11}$$

where $C$ is a coefficient approaching infinity, $\mathbb{1} \in \mathbb{R}^{n \times n}$ is a matrix with all values of 1, $\mathbf{W}_O \in \mathbb{R}^{d \times d}$ represents a trainable matrix, and CONCAT($\cdot$) denotes the concatenation operation.

## 4.3 Mutual information learning

We designed a learning strategy based on mutual information termed **Mi-Learning**, aiming to help the Ti-Transformer learn the meaningful evolution law and alleviate the over-estimation of high-degree nodes.

In the generative paradigm for dynamic link prediction task, the occurrence frequency of a node token in the sequence set can be equivalent to its degree in original graph. Thus, we can estimate the prior distribution $p(v_i)$ of node token $v_i$ by:

$$p(v_i) = \frac{\text{DEGREE}(v_i)}{\sum_{v \in \mathcal{V}} \text{DEGREE}(v)}, \tag{12}$$

where $\text{Degree}(v)$ represents the degree of node $v \in \mathcal{V}$. Then we introduce the strategy to coalesce the prior distribution into the optimization goal.

Given the sequence $\mathbf{x}^{(i)}$ in Eq.(5), the probability of the vanilla Transformer generating the $k$-th token can be formalized as:

$$p(h_k^{(i)}|\mathbf{x}_{<k}^{(i)}) = \text{Softmax}\left(\text{LayerNorm}(\mathbf{H}^{(i)}_{<k})\mathbf{W}_{\text{token}}\right), \quad (13)$$

where $h_k^{(i)}$ is the $k$-th token in $\mathbf{x}^{(i)}$, $\mathbf{x}_{<k}^{(i)}$ denotes the subsequence of $\mathbf{x}^{(i)}$ containing the first $(k-1)$ tokens, $\mathbf{H}^{(i)}_{<k}$ denotes the hidden embedding of $\mathbf{x}_{<k}^{(i)}$, and $\mathbf{W}_{\text{token}}$ is the learnable matrix to obtain the probability across the node tokens. Thus, the relation between $p(h_k^{(i)}|\mathbf{x}_{<k}^{(i)})$ and $\mathbf{H}^{(i)}_{<k}$ can be simplified as:

$$\log p(h_k^{(i)}|\mathbf{x}_{<k}^{(i)}) \sim \mathbf{H}_{<k}^{(i)} \quad (14)$$

To further explore the relationship between current token $h_k^{(i)}$ and its historical sub-sequence $\mathbf{x}_{<k}^{(i)}$, the pointwise mutual information (PMI) between them can be written as following form:

$$\log \frac{p(h_k^{(i)}, \mathbf{x}_{<k}^{(i)})}{p(h_k^{(i)})p(\mathbf{x}_{<k}^{(i)})} \sim \mathbf{H}_{<k}^{(i)}$$

$$\Longleftrightarrow \log \frac{p(h_k^{(i)}, \mathbf{x}_{<k}^{(i)})}{p(\mathbf{x}_{<k}^{(i)})} - \log p(h_k^{(i)}) \sim \mathbf{H}_{<k}^{(i)} \quad (15)$$

$$\Longleftrightarrow \log p(h_k^{(i)}|\mathbf{x}_{<k}^{(i)}) \sim \mathbf{H}_{<k}^{(i)} + \log p(h_k^{(i)}),$$

where $p(h_k^{(i)})$ is the prior distribution of token $h_k^{(i)}$. It is worth noting that the $p(h_k^{(i)})$ of node tokens can be obtained by Eq. (12) while the $p(h_k^{(i)})$ of all special tokens is set to the maximum prior probability among node tokens, i.e., $\max_{v \in \mathcal{V}} p(v)$, since that the special tokens are regarded as *Hubs* mentioned in Introduction. Without loss of generality, we multiply $\log p(h_k^{(i)})$ by a coefficient $\tau$ to adjust the importance of $\log p(h_k^{(i)})$. Thus, the Eq. (13) can be updated as:

$$p^*(h_k^{(i)}|\mathbf{x}_{<k}^{(i)}) = \text{Softmax}\left(\text{LayerNorm}(\mathbf{H}_{<k}^{(i)})\mathbf{W}_{\text{token}} + \tau \log p(h_k^{(i)})\right). \quad (16)$$

Thus, the final objective of training the model with parameters $\theta$ is defined as:

$$\mathcal{L} = -\sum_{i=1}^{|\mathcal{V}|} \sum_{k=1}^{|\mathbf{x}^{(i)}|} \log p_\theta^*(h_k^{(i)}|\mathbf{x}_{<k}^{(i)}). \quad (17)$$

## 5 EXPERIMENTS

To examine the effectiveness of our proposals, we concentrate on answering the following research questions:

**RQ1** How do our proposals perform in comparison to state-of-the-art methods on dynamic graph link prediction tasks?

**RQ2** Among our proposed components, which one contributes the most to the performance?

**RQ3** What is the impact of different loss functions on the performance of link prediction?

**RQ4** How does the choice of different mask matrices affect the performance of link prediction? Does the time-based masking scheme have an impact on the performance?

**RQ5** How is the performance of link prediction affected by different weights of the prior knowledge?

### 5.1 Experiments setup

*5.1.1 Evaluation datasets.* We utilize four real-world public datasets, including UCI [28], ML-10M [13], Hepth [18], and MMConv [19], from different domains for our experiments. To ensure the fairness of the comparative experiment and temporal alignment, we follow the previous researches [33, 40] by re-segmenting UCI, ML-10M, Hepth, and MMConv into 13, 13, 12, and 16 time periods, respectively. In addition, detailed introduction and preprocessing of four datasets are presented in Appendix A.1.

*5.1.2 Model Configurations.* Following the previous work [40], we utilize the GPT-2 model [30] as the backbone. For all the four datasets, we follow the previous setting [5, 40] by regarding the dynamic graphs as undirected graphs. The dataset is split into training, validation, and testing based on the predefined time steps mentioned in Sec.5.1.1. Specifically, the data in the first $(T-2)$ time steps are used as the training set to optimize models, the data in time period $(T-1)$ is used as the validation set to adjust the hyperparameters, and the data in the time period $T$ is used as the test set to examine the final performance.

*5.1.3 Baselines for comparison.* A number of link prediction methods have been proposed in recent years. In our modeling, we focus on dynamic graphs. Thus, we do not make comparisons with researches designed for static graphs such as GraphSage [12] and GAT-AE [37, 48]. We compare our proposals with the following baselines: (1) Two discrete-time methods, i.e., DySAT [33] and EvolveGCN [29]; (2) Six continuous-time methods, i.e., DyRep [36], JODIE [17], TGAT [42], TGN [32], TREND [39], and GraphMixer [5]; (3) A Transformer-based method, i.e., SimpleDyG [40]. Additionally, the detailed introduction of the baselines is presented in Appendix A.2.

### 5.2 Overall performance (RQ1)

To answer **RQ1**, we examine the dynamic graph link prediction performance of our proposals as well as the baselines in Table 1.

Generally, our TopDyG outperforms all the discussed baselines in all datasets. Specifically, it achieves up to a 43.27% performance improvement over the optimal baseline method on the transductive datasets including UCI, ML-10M, and MMConv. On the more challenging inductive dataset Hepth, although there is still a gap in the absolute performance values compared to the transductive scenario, both the NDCG and Jaccard achieve more than 20% improvement over the state-of-the-art baseline.

Additionally, we note an interesting phenomenon: the performance improvement of our proposal over the SOTA method is positively correlated with density of the datasets showed in Table 3. Specifically, the performance improvement of TopDyG over the SOTA methods on different datasets shows a positive correlation with the average number of triangles and the average degree for each node in the datasets. For instance, ML-10M dataset has the least number of average triangles per node among the four discussed datasets. Our proposal also has the minimal advantage against SimpleDyG, i.e., only 4.59% and 3.10% in terms of NDCG and Jaccard,

**Table 1: Performance of dynamic link prediction by our proposal and the baselines on four datasets. (In each column, the best result is bolded and the runner-up is underlined. And all the performance data of baselines is obtained from Wu et al. [40].)**

| | UCI | | ML-10M | | Hepth | | MMConv | |
|---|---|---|---|---|---|---|---|---|
| | NDCG@5 | *Jaccard* | NDCG@5 | *Jaccard* | NDCG@5 | *Jaccard* | NDCG@5 | *Jaccard* |
| DySAT | 0.010±0.003 | 0.010±0.001 | 0.058±0.073 | 0.050±0.068 | 0.007±0.002 | 0.005±0.001 | 0.102±0.085 | 0.095±0.080 |
| EvolveGCN | 0.064±0.045 | 0.032±0.026 | 0.097±0.071 | 0.092±0.067 | 0.009±0.004 | 0.007±0.002 | 0.051±0.021 | 0.032±0.017 |
| DyRep | 0.011±0.018 | 0.010±0.005 | 0.064±0.036 | 0.038±0.001 | 0.031±0.024 | 0.010±0.006 | 0.140±0.057 | 0.067±0.025 |
| JODIE | 0.022±0.023 | 0.012±0.009 | 0.059±0.016 | 0.020±0.004 | 0.031±0.021 | 0.011±0.008 | 0.041±0.016 | 0.032±0.022 |
| TGAT | 0.061±0.007 | 0.020±0.002 | 0.066±0.035 | 0.021±0.007 | 0.034±0.023 | 0.011±0.006 | 0.089±0.033 | 0.058±0.021 |
| TGN | 0.041±0.017 | 0.011±0.003 | 0.071±0.029 | 0.023±0.001 | 0.030±0.012 | 0.008±0.001 | 0.096±0.068 | 0.066±0.038 |
| TREND | 0.067±0.010 | 0.039±0.020 | 0.079±0.028 | 0.024±0.003 | 0.031±0.003 | 0.010±0.002 | 0.116±0.020 | 0.060±0.018 |
| GraphMixer | 0.104±0.013 | 0.042±0.005 | 0.081±0.033 | 0.043±0.022 | 0.011±0.008 | 0.010±0.003 | 0.172±0.029 | 0.085±0.016 |
| SimpleDyG | 0.104±0.010 | 0.092±0.014 | 0.138±0.009 | 0.131±0.008 | 0.035±0.014 | 0.013±0.006 | 0.184±0.012 | 0.169±0.010 |
| **TopDyG** (Ours) | **0.149±0.007** | **0.101±0.006** | **0.144±0.002** | **0.135±0.002** | **0.042±0.004** | **0.017±0.002** | **0.242±0.040** | **0.218±0.030** |
| Improvement | 43.27% | 9.70% | 4.59% | 3.10% | 21.14% | 27.31% | 31.64% | 28.75% |

respectively. On the other hand, MMConv dataset is a bipartite dynamic graph without triangle, but our TopDyG still has about 30% improvement. In view of the above phenomena, we analyze statistics of MMConv showed in Table 3 and consider that although it misses the triangle pattern, it has an average node degree greater than 26, the largest one in the four datasets. In such case, our topology injection method not only does not fail, but also assist the model to pay more attention to the relationship between neighbor nodes and the ego node in the sequences of subgraphs, rather than to the relationship with special tokens. It can be attributed to the fact that the average number of triangles and node degrees are positively correlated with density of the datasets. To sum up, the more dense the graphs, the more obvious the topological features contribute to the performance of TopDyG.

### 5.3 Ablation study (RQ2)

To answer **RQ2**, we perform an ablation study to get a deep insight into each component of our proposed TopDyG model. We remove or replace a certain component of TopDyG and examine the corresponding performance of the incomplete TopDyG on all discussed datasets, which is denoted as the notation 'w/o'. In particular, there are two individual TopDyG models we want to investigate, i.e., $\text{TopDyG}_{w/o[Ti]}$ that removes the topology-injected mechanism, and $\text{TopDyG}_{w/o[Mi]}$ that removes the mutual information learning strategy. We compute the relative performance change rates of incomplete TopDyG models against the integrated TopDyG model. Each ablation result is computed with the average of relative change rates for the NDCG and Jaccard indices. We report the detail results in Table 2.

As shown in Table 2, removing either of the two components results in an obvious performance drop under different circumstances. Specifically, "w/o Ti-Transformer" causes the biggest drop of both NDCG and Jaccard on Hepth and MMconv datasets. Performance diminish on Hepth indicates that modeling the topology feature is an effective way to boost the dynamic graph link prediction under the inductive setting. Furthermore, the performance degradation of the model on different datasets after removing topology feature

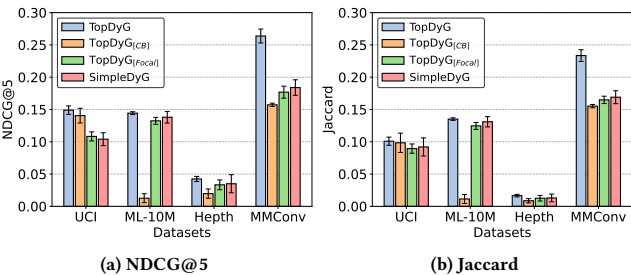

(a) NDCG@5    (b) Jaccard

**Figure 4: Performance comparison to examine the impact of loss functions.**

aligns with the trend observed in Table 1. That is, the greater the average number of triangles and the average degree of the dataset, the more the performance is influenced by topology features, which precisely corroborates our perspective presented in Sec 5.2.

In addition, "w/o Mi-Learning" leads to the biggest drop of both NDCG and Jaccard on UCI and ML-10M datasets. Performance drop on the two datasets indicates that leveraging the prior distribution in Mi-Learning is an effective way to improve the model capability under transductive setting. Overall, the impact of removing different modules on transductive datasets is similar while topological features have a greater influence on Hepth than other three datasets, indicating that priority should be given to considering the topological structure of dynamic graphs in inductive scenarios.

### 5.4 Impact of optimization goals (RQ3)

To answer **RQ3** to see the impact of the loss functions for dynamic link prediction task, we compare the performance of our proposals and several variants, i.e., $\text{TopDyG}_{[CB]}$ and $\text{TopDyG}_{[Focal]}$. Specifically, $\text{TopDyG}_{[CB]}$ and $\text{TopDyG}_{[Focal]}$ denote the Mi-Learning module in TopDyG is replaced by the *Class Balanced Loss* [6] and *Focal Loss* [20], respectively. Such methods are utilized for alleviating the class imbalanced problem.

First, we concentrate on the comparison between TopDyG based methods and the SOTA baseline SimpleDyG. As shown in Fig. 4,

**Table 2: Ablation studies of TopDyG for the link prediction tasks. The biggest drop in each column is appended ↓.**

| | UCI | | ML-10M | | Hepth | | MMConv | |
|---|---|---|---|---|---|---|---|---|
| | NDCG@5 | *Jaccard* | NDCG@5 | *Jaccard* | NDCG@5 | *Jaccard* | NDCG@5 | *Jaccard* |
| TopDyG | **0.149±0.007** | **0.101±0.006** | **0.144±0.002** | **0.135±0.002** | **0.042±0.004** | **0.017±0.002** | **0.242±0.040** | **0.218±0.030** |
| w/o Ti-Transformer | 0.124±0.011 -16.48% | 0.093±0.005 -8.33% | 0.137±0.007 -5.43% | 0.127±0.006 -6.22% | 0.025±0.006 -40.19% ↓ | 0.010±0.002 -37.76% ↓ | 0.164±0.003 -32.29% ↓ | 0.156±0.002 -28.26% ↓ |
| w/o Mi-Learning | 0.110±0.011 -25.95% ↓ | 0.088±0.015 -12.71% ↓ | 0.133±0.014 -7.93% ↓ | 0.125±0.013 -7.48% ↓ | 0.036±0.017 -15.47% | 0.013±0.007 -20.00% | 0.168±0.010 -30.85% | 0.158±0.012 -27.25% |

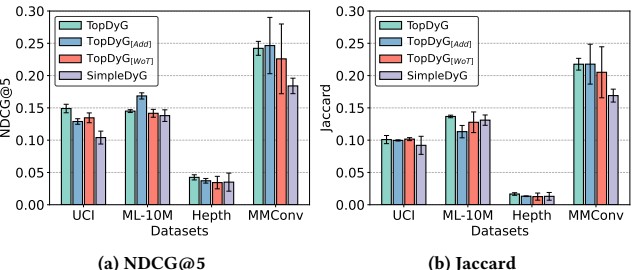

(a) NDCG@5      (b) Jaccard

**Figure 5: Performance comparison to examine the impact of topology injection techniques.**

TopDyG$_{[Focal]}$ only outperforms SimpleDyG on UCI datasets while other variants of TopDyG obtains worse performance than SimpleDyG on all datasets. We consider that such phenomenon is caused by the reweighting mechanism in Focal Loss, aiming to force the model pay attention to the difficult samples in classification tasks. However, token prediction is more challenging than sample classification due to the vast number of tokens from which to choose when generating the next token. Consequently, methods designed for addressing long-tail classification are not directly applicable to generative tasks. Such inapplicability is highlighted by the TopDyG$_{[CB]}$ on ML-10M dataset, where the TopDyG$_{[CB]}$ obtains the worst performance among the discussed models.

Next we zoom in on the performance of TopDyG. It is obvious that our TopDyG maintains its advantage compared with several variants. Specifically, not only does it have the best average performance in terms of NDCG and Jaccard, but it also exhibits the smallest standard deviation, illustrating the highest stability among all the methods discussed. Therefore, we consider that modeling the mutual information serves as a straightforward and effective approach to boost the performance of the Transformer-based model with real-world graphs.

## 5.5 Effect of topology injection methods (RQ4)

To investigate the effect of different types of the topology injection technique, we modify the adjacent matrix $A$ of input sequence **x** into two variants, i.e. the topological variant $A_{[Add]}$ and temporal variant $A_{[WoT]}$. The results are shown in Fig. 5.

From the topological perspective, we accumulate the number of times each edge appears in the subgraph and use it as an element in the adjacency matrix:

$$A_{[Add]pq} = \text{Count}([p, q]), \tag{18}$$

where $\text{Count}([p, q])$ denotes the frequency of the edge $(p, q)$ appears in the dynamic graph. From the temporal perspective, we consider that the importance of historical edges diminishes over time. Thus, the corresponding matrix $A_{[WoT]}$ is defined as:

$$A_{[WoT]pq} = \sum_{m=1}^{T} \exp(m - T) \times g([p, q], m), \tag{19}$$

where $WoT$ is the abbreviation of "weighting over time", $g([p, q], m) = \{0, 1\}$ is a function discriminating the existence of edge $(p, q)$ at time period $m$.

We can observe that almost all the TopDyG-based models outperform the SimpleDyG on all discussed datasets. However, for ML-10M dataset, TopDyG$_{[Add]}$ obtains the best performance in terms of NDCG while the worst in terms of Jaccard. It can be attributed to the accumulating mechanism since that it overemphasizes the importance of nodes with high degrees, leading the model to repeatedly generate such high-frequency nodes. Therefore, the diversity of prediction decreases and the Jaccard metric is reduced. For the TopDyG$_{[WoT]}$, it outperforms TopDyG$_{[Add]}$ in terms of the Jaccard metric for most datasets. This is mainly because that it generally reduces the weight of high-degree nodes compared to TopDyG$_{[Add]}$, mitigating the model's preference for high-degree nodes. On the other hand, either TopDyG$_{[Add]}$ or TopDyG$_{[Add]}$ has the worst stability on most of the datasets, indicating that the re-weighting operation is not the best choice to assist the model capture the features.

## 5.6 Impact of the prior knowledge (RQ5)

In order to investigate the impact of the weight of prior knowledge in Mi-Learning, we test the performance of TopDyG on all datasets by ranging the weight $\tau$ from 0 to 1.5 with a step size of 0.25. The results are shown in Fig. 6.

The metrics, i.e., NDCG and Jaccard, reach the peaks when $\tau$ is close to 0.75, showing an overall upward trend from 0 to 0.75, and a general downward trend from 0.75 to 1.5. When $\tau = 0$ or $\tau = 1.5$, TopDyG shows the worst performance on all the four datasets. Specifically, $\tau = 0$ means that the optimization goal degrades into the conditional probability without prior knowledge of the dataset. Thus, the model cannot be trained well to capture the evolution pattern, and it is bothered by the imbalanced distribution of real-world graphs.

On the other hand, setting the value of $\tau$ to 1.5 means that the weight of the prior distribution is overemphasized in the optimization goal, which leads to an excessive averaging of node importance,



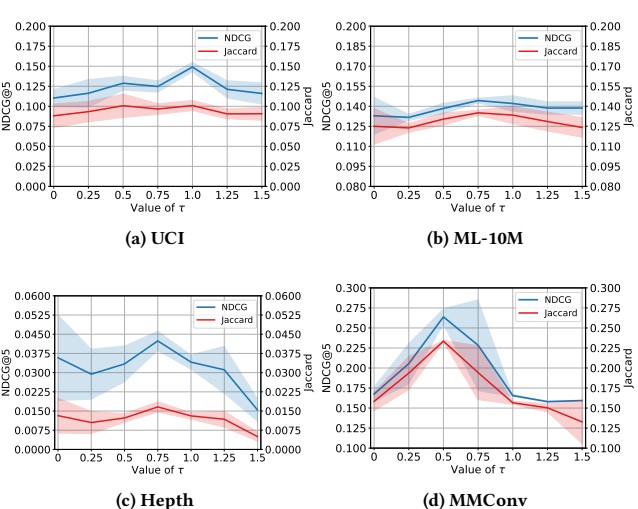

(a) UCI

(b) ML-10M

(c) Hepth

(d) MMConv

**Figure 6: Performance of TopDyG on four datasets with the parameter $\tau$ in Eq. (16) changing from 0 to 1.5. The dark line represents the mean, and the light shaded area indicates the standard deviation.**

preventing the model from identifying important nodes. Furthermore, the phenomenon is particularly pronounced in the MMConv dataset because it is a bipartite graph lacking triangles, with the 1-hop subgraph exhibiting a single radial pattern. Therefore, over-averaging the degree features of nodes leads to impaired structural features and the worst performance.

## 5.7 Case study

In this subsection, we show an example from the test set of UCI to illustrate the different attention matrices produced by TopDyG and the state-of-the-art baseline SimpleDyG; see Fig. 7.

Fig. 7a shows the normalized adjacency of a 1-hop subgraph sequence with node 848 as the center, where the sequences on the left and bottom sides are the original input sequences. The darker the area in the plot, the more important the information is. It can be observed from the Fig. 7b and 7c that SimpleDyG roughly focuses its attention on the first few tokens without distinguishing between special tokens and node tokens, whereas TopDyG directs its attention primarily to the central node token. Additionally, as shown in Fig. 7d and 7e, TopDyG can also pay attention to the relationships between neighbors of the center node, while SimpleDyG is poor at modeling such structural features in the input sequence. In summary, compared to the attention matrices produced by SimpleDyG, those produced by TopDyG are more similar to the original topology shown in Fig. 7a. This confirms that TopDyG has a better capability for modeling the inner relationships of neighboring nodes than SimpleDyG.

## 6 CONCLUSION

In this paper, we propose a topology-aware Transformer architecture for dynamic graph link prediction (TopDyG), which consists of a topology-injected Transformer (Ti-Transformer) and a mutual information learning strategy (Mi-Learning). Specifically, Ti-Transformer explores the relationship between the neighboring

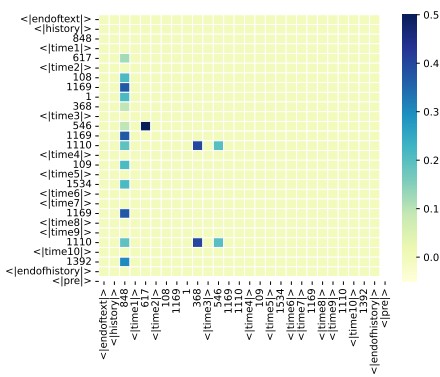

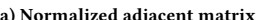

(a) Normalized adjacent matrix

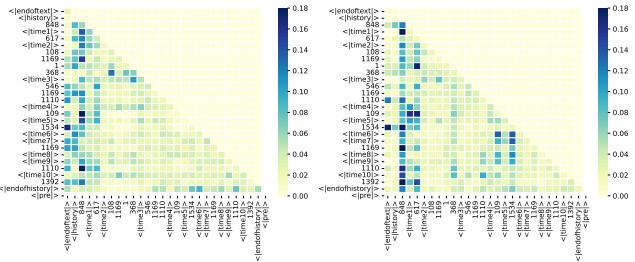

(b) Attention of head 0 in SimpleDyG    (c) Attention of head 0 in TopDyG

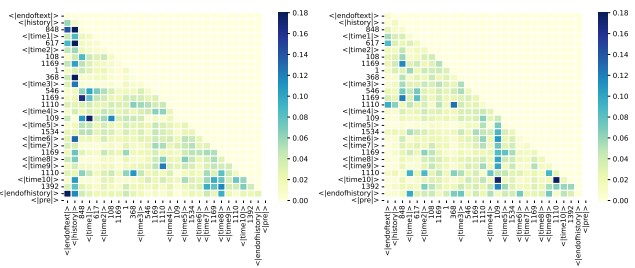

(d) Attention of head 6 in SimpleDyG    (e) Attention of head 6 in TopDyG

**Figure 7: An example of attention matrices generated by TopDyG and SimpleDyG.**

nodes of the center node in a serialized 1-hop subgraph, and feeds the topological feature into Transformer model in an explicit way without extra learnable modules. Additionally, Mi-Learning models the mutual information between nodes in a 1-hop subgraph, alleviating the over-estimation on high-degree nodes in real-world graphs with the prior knowledge of them. Experimental results on four real-world public datasets exhibit the advantages of TopDyG on improving the performance of dynamic link prediction in terms of NDCG and Jaccard. Moreover, we find the superiority of TopDyG is obvious when dealing with graphs with high density.

As for future work, we plan to investigate how to eliminate the effects of the imbalanced distribution and reveal the significance of structural features in an explainable way. Besides, we are interested in exploring the application of our method in combining large language models (LLMs) with graph data. Thus, we can harness the understanding and generation capabilities of LLMs to facilitate the completion of graph-related tasks.

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

# A APPENDIX

## A.1 Datasets

**Table 3: Statistics of the datasets used in our experiments**

| Dataset | UCI | ML-10M | Hepth | MMConv |
|---|---|---|---|---|
| Domain | Social | Rating | Citation | Conversation |
| Paradigm | Transductive | Transductive | Inductive | Transductive |
| # Nodes | 1,781 | 15,841 | 4,737 | 7,415 |
| # Edges | 16,743 | 48,561 | 14,831 | 91,986 |
| # Density | 0.0105 | $3.87 \times 10^{-4}$ | 0.0013 | 0.0033 |
| # Triangles | 13,580 | 15,962 | 11,328 | 0 |
| # Avg. Triangles | 7.6249 | 1.0076 | 2.3914 | 0 |
| # Avg. Degree | 18.8020 | 6.1311 | 6.2618 | 26.2980 |
| # Time Periods | 13 | 13 | 12 | 16 |

- **UCI** is a social network dataset aiming at sustaining interaction among students at University of California, Irvine and help them enlarge their circles of friends.
- **ML-10M** is collected from the MovieLens website. It consists of user-tag interactions, where the edges represent interactions, nodes denote users and tags.
- **Hepth** is from the e-print arXiv and covers scientific collaborations between authors papers submitted to High Energy Physics - Theory category. It is worth noting that Hepth contains new emerging nodes as time goes on.
- **MMConv** is a multi-modal conversational dataset, a fully annotated collection of human-to-human role-playing dialogues spanning over multiple domains and tasks.

For Hepth dataset where text is used as node attributes, we align with previous work [40] by employing the word2vec model [23] to encode the text into embedding vectors. The detailed statistics of the four datasets after preprocessing are provided in Table. 3, where Avg. Triangle represents the average number of triangles that can be formed directly with the nodes that are 1-hop away from each node.

## A.2 Baseline for comparison

- **DySAT** [33] is a neural architecture learning node representations to capture dynamic graph structural evolution along the two dimensions of structural neighborhood and temporal dynamics.
- **EvolveGCN** [29] captures the dynamism of the graph sequence with an RNN to evolve the GCN parameters rather than resorting to node representations.
- **DyRep** [36] is a two-time scale deep temporal point process model that captures the interleaved dynamics of the observed processes with an unsupervised procedure.
- **JODIE** [17] is a coupled recurrent neural network model, which employs two RNNs to update the embeddings at every corresponding interaction.
- **TGAT** [42] contains the self-attention mechanism as building block and a novel functional time encoding technique based on the classical Bochner's theorem from harmonic analysis.
- **TGN** [32] is a generic, efficient framework for deep learning on dynamic graphs represented as sequences of timed events.
- **TREND** [39] is a framework for temporal graph representation learning, driven by temporal event and node dynamics and built upon a Hawkes process-based GNN.
- **GraphMixer** [5] employs MLPs and neighbor mean-pooling to summarize the temporal links and node representation information, respectively.
- **SimpleDyG** [40] is a Transformer-based simple approach for modeling dynamic graphs with time alignment technique.

