# OpenReview forum: "Triangle Matters! TopDyG: Topology-aware Transformer for Link Prediction on Dynamic Graphs"
_ACM.org/TheWebConf/2025/Conference — WWW 2025 Poster_

### Official Review · Reviewer_uwgS · 2024-11-23

**Novelty:** 4
**Technical Quality:** 5

**Review:**

**Summary**

The paper proposes a Topology-aware Transformer on Dynamic Graphs (TopDyG) for link prediction, addressing the challenges of modeling fine-grained structures and avoiding over-estimation of high-degree nodes. The approach incorporates a Ti-Transformer for capturing topological features and Mi-Learning for mitigating node imbalance, demonstrating superior performance on four public datasets, particularly in high-density graphs.

**Strengths**

1. The proposed TopDyG model addresses key challenges in dynamic graph link prediction by introducing a topology-aware Transformer and mutual information learning

2. The experimental evaluation demonstrates the superiority of TopDyG over state-of-the-art methods, with improvements in NDCG and Jaccard scores across various datasets.



**Weaknesses**

1. The way topology is added to the Transformer is too basic. It may not fully use the global structure of the graph.

2. The approach may not be fully scalable to larger graphs with more complex topologies. The paper does not provide sufficient discussion on the scalability of TopDyG when applied to large-scale dynamic networks.

3. The model heavily relies on degree distribution to address high-degree nodes. This approach might cause the model to undervalue the importance of low-degree nodes in graphs with highly uneven degree distributions.

**Questions:**

What are the time complexity and efficiency of the algorithm

**Reviewer Confidence:**

3: The reviewer is confident but not certain that the evaluation is correct

**Scope:**

3: The work is somewhat relevant to the Web and to the track, and is of narrow interest to a sub-community

---

### Official Review · Reviewer_ENxr · 2024-11-29

**Novelty:** 5
**Technical Quality:** 5

**Review:**

**Paper Summary:**

The paper addresses the problem of dynamic graph link prediction, particularly in real-world scenarios such as social networks, citation networks, and recommendation systems. Existing Transformer-based methods in dynamic graph prediction have two main issues: (1) the inability to effectively model fine-grained graph structures (such as triangles) during graph serialization, and (2) the over-estimation of high-degree nodes.

**Summary of Strengths:**
+ The paper introduces a novel method that combines the topology-aware Ti-Transformer and Mi-Learning to tackle the key issues in dynamic graph link prediction, namely graph serialization and the over-estimation of high-degree nodes.

+ The results demonstrate significant improvements in performance (NDCG and Jaccard) compared to state-of-the-art methods, particularly on high-density graphs.


**Summary of Weaknesses:**

+ Although the topology-injected Transformer offers improvements, the model might be computationally more complex than simpler methods. The trade-off between performance and computational efficiency is not discussed in detail.

**Questions:**

In addition to performance, computational efficiency is an important aspect of evaluating a method. How does TopDyG compare to existing methods in terms of training and inference time? In practical applications, can the model maintain efficiency when handling large dynamic graphs?

**Reviewer Confidence:**

2: The reviewer is willing to defend the evaluation, but it is likely that the reviewer did not understand parts of the paper

**Scope:**

3: The work is somewhat relevant to the Web and to the track, and is of narrow interest to a sub-community

---

### Official Review · Reviewer_miDj · 2024-12-01

**Novelty:** 4
**Technical Quality:** 4

**Review:**

This paper proposes TopDyG, a topology-aware Transformer architecture for link prediction on dynamic graphs. TopDyG addresses two limitations of existing Transformer-based methods: (1) loss of triangle information during graph serialization and (2) over-estimation of high-degree nodes. It introduces two key components: (1) Ti-Transformer, which captures explicit topological features using normalized adjacency matrices without additional trainable modules, and (2) Mi-Learning, which leverages mutual information with prior knowledge to mitigate the over-estimation of high-degree nodes. Experiments on four real-world datasets demonstrate the effectiveness of TopDyG.

**Pros:**
1. TopDyG offers a unique perspective on incorporating topology information into Transformers, addressing a critical limitation of existing methods.
2. The Ti-Transformer and Mi-Learning components are straightforward and efficient, requiring minimal additional computational resources.
3. The paper is well-structured and clearly written, making it easy to understand the proposed approach and its contributions.

**Cons:**
1. While the paper demonstrates the effectiveness of TopDyG, it could benefit from a deeper exploration of the theoretical reasons behind its success.
2. While the additional computation cost of TopDyG is minimal, a more detailed analysis of its time and space complexity compared to other methods would be beneficial.
3. The paper primarily focuses on the structural features of the graph. While the Ti-Transformer does capture some temporal information by using normalized adjacency matrices, a more explicit treatment of temporal dynamics, such as incorporating temporal features or using time-encoded embeddings, could potentially improve performance.
4. The Mi-Learning component relies on the distribution of node degrees as prior knowledge. While this is a common approach, it may introduce biases if the real-world graph has a different degree distribution. Exploring alternative ways to obtain prior knowledge or incorporating additional features could help mitigate these biases.

**Questions:**

1. How does TopDyG perform on dynamic graphs with different types of topologies, such as scale-free networks or community structures?
2. Can the Ti-Transformer be extended to capture higher-order topological features, such as cliques or motifs?

**Reviewer Confidence:**

3: The reviewer is confident but not certain that the evaluation is correct

**Scope:**

3: The work is somewhat relevant to the Web and to the track, and is of narrow interest to a sub-community

---

### Official Review · Reviewer_1qfe · 2024-12-02

**Novelty:** 5
**Technical Quality:** 6

**Review:**

This paper addresses the important problem of link prediction in dynamic graphs by proposing TopDyG, a novel Transformer-based architecture that explicitly incorporates topology information. The key innovations are: (1) A topology-injected Transformer that captures triangle relationships in serialized graph sequences, and (2) A mutual information learning strategy to mitigate over-emphasis on high-degree nodes.

For transformers, the key insight is that vanilla Transformers lose important triangle relationships when graphs are serialized into sequences. The paper overcomes this drawback by injecting the normalized adjacency matrix into transformer's attention mask. This allows the model to maintain awareness of the original graph structure while processing the sequence.

The mutual information strategy addresses the problem of over-emphasizing high-degree nodes (hubs) in scale-free networks, which many real-world networks exhibit. It works by modifying the optimization objective to include mutual information between current token and its history, which helps he model learn intrinsic patterns rather than just focusing on high-degree nodes, which vanilla Transformers tend to do.

The problem studied in the paper is well-motivated as using transformer to perform downstream machine learning tasks on graphs is a popular topic in social network analysis. The method of the paper shows strong performance and significant improvement over the previous state-of-the-art. It would be great if the author could add more discussion on how much more computational complexity the new approach introduces.

**Questions:**

None

**Reviewer Confidence:**

3: The reviewer is confident but not certain that the evaluation is correct

**Scope:**

4: The work is relevant to the Web and to the track, and is of broad interest to the community

---

### Official Review · Reviewer_MMAa · 2024-12-04

**Novelty:** 5
**Technical Quality:** 5

**Review:**

The paper presents TopDyG, a topology-aware Transformer architecture designed for link prediction on dynamic graphs. The model incorporates a topology-injected Transformer (Ti-Transformer) to capture structural features, particularly triangles, and a mutual information learning (Mi-Learning) strategy to mitigate the over-estimation of high-degree hub nodes.

**Questions:**

1. The core contribution of this paper, the Ti-Transformer, demonstrates significant performance improvements, yet it does not introduce fundamentally new concepts or techniques. Instead, it primarily leverages the structural information provided by the Temporal Ego-graph. What is the significant difference in subgraph extraction between the TopDyG and the SimpleDyG?
2. The datasets used in this study are relatively limited, and there is a lack of discussion regarding the scalability of the TopDyG model. How does the model's performance scale with larger, more complex graphs, and what are the computational requirements?
3. The paper claims improvements on dense graphs, but does the model generalize well to sparse graphs or graphs with different characteristics?
4. Different topological injection methods, such as TopDyG[Add] and TopDyG[WoT], show instability only on the MMConv dataset. What could be the underlying reasons for this?
5. The paper claims that incorporating node degree as a prior distribution mitigates the over-estimation issue for high-degree nodes. Could you provide more intrinsic theoretical proof or more robust experimental validations?

**Reviewer Confidence:**

4: The reviewer is certain that the evaluation is correct and very familiar with the relevant literature

**Scope:**

3: The work is somewhat relevant to the Web and to the track, and is of narrow interest to a sub-community